# General molten-salt route to three-dimensional porous transition metal nitrides as sensitive and stable Raman substrates

Haomin Guan[1,2], Wentao Li[1], Jing Han[3], Wencai Yi [4], Hua Bai[1], Qinghong Kong[2] & Guangcheng Xi [1✉]

Transition metal nitrides have been widely studied due to their high electrical conductivity and excellent chemical stability. However, their preparation traditionally requires harsh conditions because of the ultrahigh activation energy barrier they need to cross in nucleation. Herein, we report three-dimensional porous VN, MoN, WN, and TiN with high surface area and porosity that are prepared by a general and mild molten-salt route. Trace water is found to be a key factor for the formation of these porous transition metal nitrides. The porous transition metal nitrides show hydrophobic surface and can adsorb a series of organic compounds with high capacity. Among them, the porous VN shows strong surface plasmon resonance, high conductivity, and a remarkable photothermal conversion efficiency. As a new type of corrosion- and radiation-resistant surface-enhanced Raman scattering substrate, the porous VN exhibits an ultrasensitive detection limit of $10^{-11}$ M for polychlorophenol.

[1] Institute of Industrial and Consumer Product Safety, Chinese Academy of Inspection and Quarantine, Beijing, P. R. China. [2] School of the Environment and Safety Engineering, Jiangsu University, Zhenjiang, P. R. China. [3] Technical Center of Qianjiang Customs House, General Administration of Customs, Hangzhou, P. R. China. [4] School of Physics and Physical Engineering, Qufu Normal University, Qufu, P. R. China. ✉email: xiguangcheng@caiq.org.cn

Metallic transition metal nitrides (TMNs) represented by molybdenum nitride (MoN), tungsten nitride (WN), vanadium nitride (VN), etc. have electronic structures similar to those of platinum-group elements, and therefore exhibit remarkable catalysis, sensing and energy storage properties, and have shown considerable advantage in reducing costs[1–5]. Moreover, TMNs have high electrical conductivity, extremely high mechanical strength, corrosion resistance, and oxidation resistance, so they have attracted more and more research interest in several application fields[6–10]. For examples, two-dimensional (2D) $Ti_2N$ nanosheets have been reported to be an excellent surface-enhanced Raman scattering (SERS) substrate[11]. Hussain et al. demonstrated that plasmonic TiN is an efficient hybrid photo-detector under low light conditions[12]. More recently, Schramke et al. reported that plasmonic TMN nanocrystals with resonances at near-Infrared wavelengths have strong photothermal effects[13]. However, because the activation energy barrier to be overcome during their nucleation and growth process is considerable high, the preparation of crystalline TMNs generally requires extremely high temperature (>1000 °C) and pressure (several GPa)[14,15]. Therefore, looking for a mild preparation route has always been the pursued objective of chemists and material scientists.

With the untiring efforts, researchers have recently succeeded in preparing metallic TMNs under atmospheric pressure by improved gas–solid reaction routes[16–19]. For example, Zhou and Gogotsi et al. reported salt-templated synthesis of 2D TMN nanostructures[17,18]. Liu et al. prepared TMNs architectures by using diatomite as template[19]. Although several interesting TMN materials have been prepared by the newly developed methods, the reaction temperature is still as high as 650–900 °C. In addition to increasing energy consumption and operation difficulty, the high-temperature preparation methods are generally difficult to use for the controlled preparation of nanostructured materials with high surface area and porosity, such as micron-scale three-dimensional (3D) porous structures, because large-volume porous microparticles with fine pore structures tend to sinter and collapse at high temperatures.

Molten-salt reaction route has been proved to be a very effective synthesis route[20–23], but there is no low-temperature molten-salt synthesis method for TMNs with highly porous structure. Herein, we report that metallic TMNs (VN, MoN, WN, TiN) with large surface area and porosity are prepared at 290 °C and 1 atm by a mild molten-salt method. Environmentally friendly $ZnCl_2$ is used as the molten-salt phase, and no volatile solvent is used. The TMNs show micron-scale 3D macrostructures and contain rich microstructures ranging from mesopores to macropores, with high specific surface area of 103.7–156.8 $m^2 g^{-1}$ and pore volume of 0.26–0.81 $cm^3 g^{-1}$. Interestingly, trace water, which is usually considered to be an unfavorable factor in molten-salt reactions, is found to be a key factor for the formation of these 3D porous TMNs. This unique structure gives them excellent hydrophobic properties and can adsorb a serious of organic compounds with high capacity. Among them, 3D porous VN are found to have a strong plasma resonance effect and outstanding photothermal conversion efficiency (67.3%). As a new type of corrosion- and radiation-resistant SERS substrate, the 3D porous VN exhibits an ultrasensitive detection limit of $10^{-11}$ M for environmental pollutants.

## Results

### Synthesis and characterization of 3D porous VN. As shown in Fig. 1a, the TMNs with large surface area and porosity are synthesized by a mild and simple molten-salt method. We use the preparation process of 3D porous VN to illustrate this low-temperature molten-salt strategy. In the traditional methods for synthesizing TMNs, $NH_3$ is generally used as the nitrogen source.

In order to activate $NH_3$, considerable high reaction temperature and pressure are required. In order to reduce the reaction barrier, in the current low-temperature molten-salt route, chemically active $Li_3N$ is used as a nitrogen source, and $VCl_3$ is used as a vanadium source. Due to the suitable melting point (~285 °C), lower price, and environmental friendliness, $ZnCl_2$ and $ZnCl_2 \cdot 6H_2O$ (crystal water is very important for the formation of 3D porous structure) are used as the molten-salt system. It should be noted that, unlike the solvent thermal route of benzene used as a solvent reported by Qian et al.[24], no toxic and volatile solvent is used in the current method, which greatly reduces the environmental damage during the production process. After washing and drying, blue–black products were obtained (Fig. 1b).

X-ray powder diffraction (XRD) is used to detect the crystal phase of the blue–black products. As shown in Supplementary Fig. 1, all XRD diffraction peaks can be indexed as a cubic-phase VN (JCPDS No. 35-0768), and no other crystalline impurities have been detected, which shows that pure and crystalline VN has been prepared by this low-temperature molten-salt method. X-ray photoelectron spectroscopy (XPS) is used to determine the chemical state of the elemental vanadium in the sample. The fine XPS spectrum of V-2p shows that in addition to the predominant $V^{3+}$ ions, there are a small amount of $V^{4+}$ and $V^{5+}$ ions in the sample (Supplementary Fig. 2). These vanadium ions with high valence should be derived from the surface oxidized vanadium species, such as $VO_2$ and $V_2O_5$.

Scanning electron microscope (SEM) and transmission electron microscope (TEM) are used to characterize the morphology and microstructure of the VN samples. First, a large range of low-magnification SEM image shows that the samples are composed of a large number of micron-sized particles, which are mainly columnar in shape (Fig. 1c). The enlarged SEM image shows that the columnar microparticles are not solid, but are composed of a large number of fibrous-like and vesicle-like nanostructures (Fig. 1c, d and Supplementary Fig. 3). We randomly inspected multiple columnar microparticles and confirmed that they all consisted of similar porous microstructures (Supplementary Fig. 4). TEM images further confirmed the 3D porous characteristics of such particles (Fig. 1e, f). In such microparticles, the fibrous structure and the vesicle-like structure are interconnected to form a mesoporous–macroporous composite porous structure together. Interestingly, the TEM image shows that the fibrous-like structures are not common solid structures, but rather hollow, very similar to nanotube structures. This nanotube feature is particularly clear from the sample's high-angle annular dark field (HAADF) image (Fig. 1g). To our knowledge, this may be the first time to obtain a nanotube structure of cubic-phase VN. The wall thickness of the nanotubes is only about 6–8 nm (Supplementary Fig. 5). Furthermore, the high-resolution transmission electron microscope (HRTEM) image reveals that the nanotubes are highly crystalline, and the lattice fringe spacing of 0.24 nm corresponds to the VN (111) (Fig. 1h). In addition to the 3D columnar porous structures, the VN samples also contain some ellipsoidal 3D structures. SEM and TEM characterizations confirmed that these ellipsoidal particles also have obvious 3D porous structures (Supplementary Fig. 6). The element mapping recorded from energy dispersion spectrum (EDS) proves that vanadium and nitrogen are evenly distributed throughout the microparticles (Fig. 1i–k). Moreover, the EDS spectrum of the sample shows that the atomic ratio of vanadium and nitrogen of the sample species is about 1.06, which is very close to the stoichiometric ratio of vanadium nitride (Fig. 1l). In addition, since the signals of oxygen (O K, 0.52 KeV) and vanadium (V L, 0.51 KeV) have some overlap, in order to further confirm the composition of the sample, the X-ray fluorescence component characterization technology is adopted, which showed that the

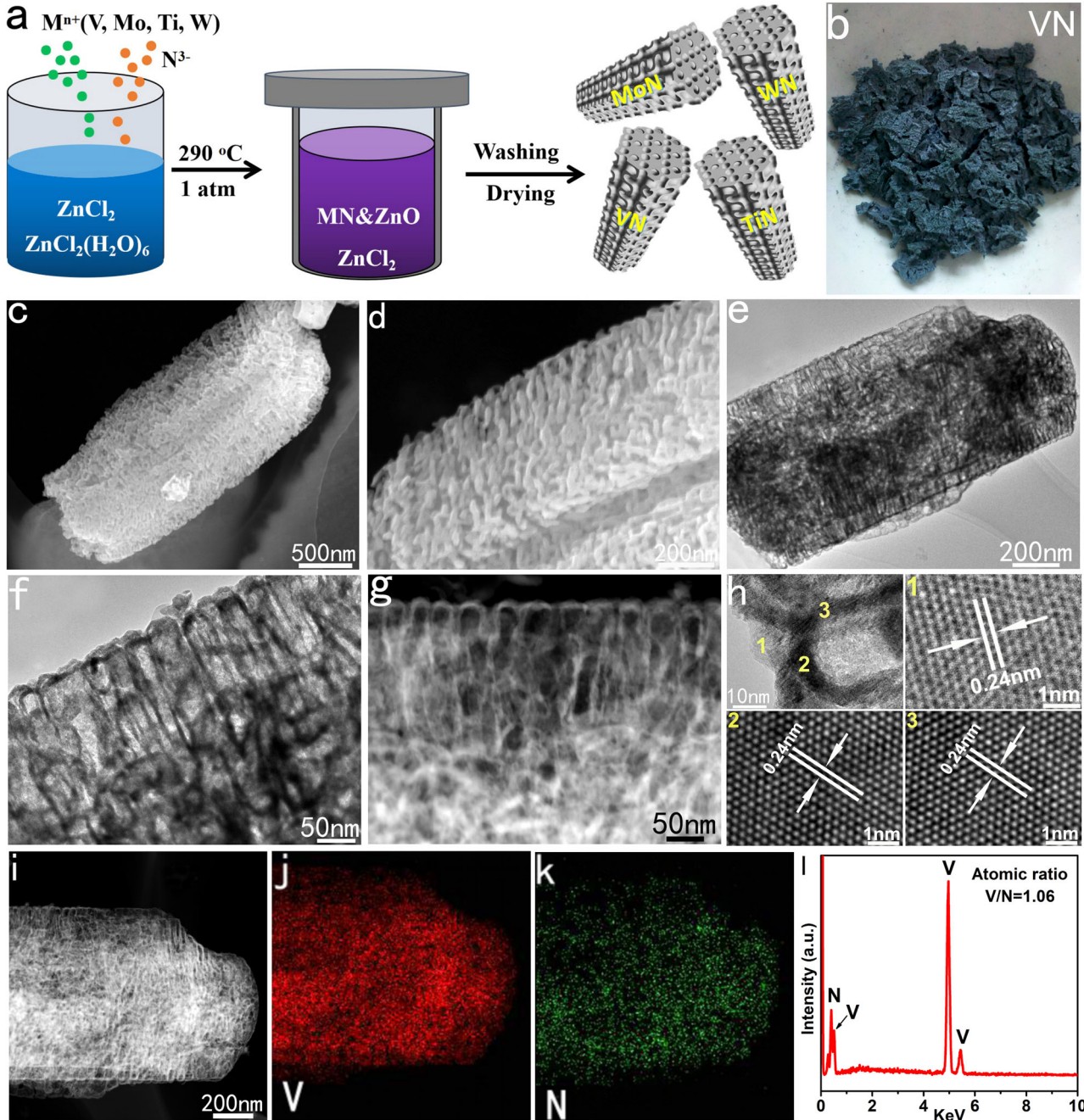

**Fig. 1 Synthesis and characterization of the 3D porous VN. a** Schematics illustrating the synthesis of highly crystalline 3D porous TMNs. **b** The obtained VN sample. **c, d** SEM images of VN. **e, f** TEM images of VN. **g** HAADF image of VN. **h** HRTEM image of VN. **i-l** EDS mapping and spectrum of VN.

atomic ratio of V/N is 1.09, which is highly consistent with their EDS characterization results.

**Universality of the method**. Using this mild molten-salt method, we not only produced the micron-scale 3D porous VN, but also prepared MoN, WN, and TiN with similar structures, which confirms that the current method has certain universality for the synthesis of 3D porous TMNs. Supplementary Figure 7 shows the XRD patterns of the prepared MoN, WN, and TiN samples, which can be respectively indexed as hexagonal phase MoN (JCPDS No. 089-5024), cubic-phase WN (JCPDS No. 65-2898), and cubic-phase TiN (JCPDS No. 38-1420). XPS characterization results show that the oxidation states of the three TMNs really belong to MoN, WN, and TiN (Supplementary Fig. 8). SEM,

TEM, and HAADF images (Fig. 2a–i) show that the obtained MoN, WN, and TiN products have a morphology and microstructure very similar to that of 3D porous VN. Moreover, the HRTEM images (inserted in Fig. 2c, i) demonstrated that the as-synthesized 3D porous MoN, WN, and TiN have high crystallinity. In addition, the EDS spectra of the samples show that the atomic ratio of Mo/N, W/N, and Ti/N are about 1.08, 1.05, and 1.04 (Supplementary Fig. 9). These results prove that this low-temperature molten-salt method is a general method for the synthesis of metallic 3D porous TMNs.

**Formation mechanism of the 3D porous TMNs**. In order to understand the formation mechanism of this 3D porous TMNs, we used VN as an example to explore the intermediate products

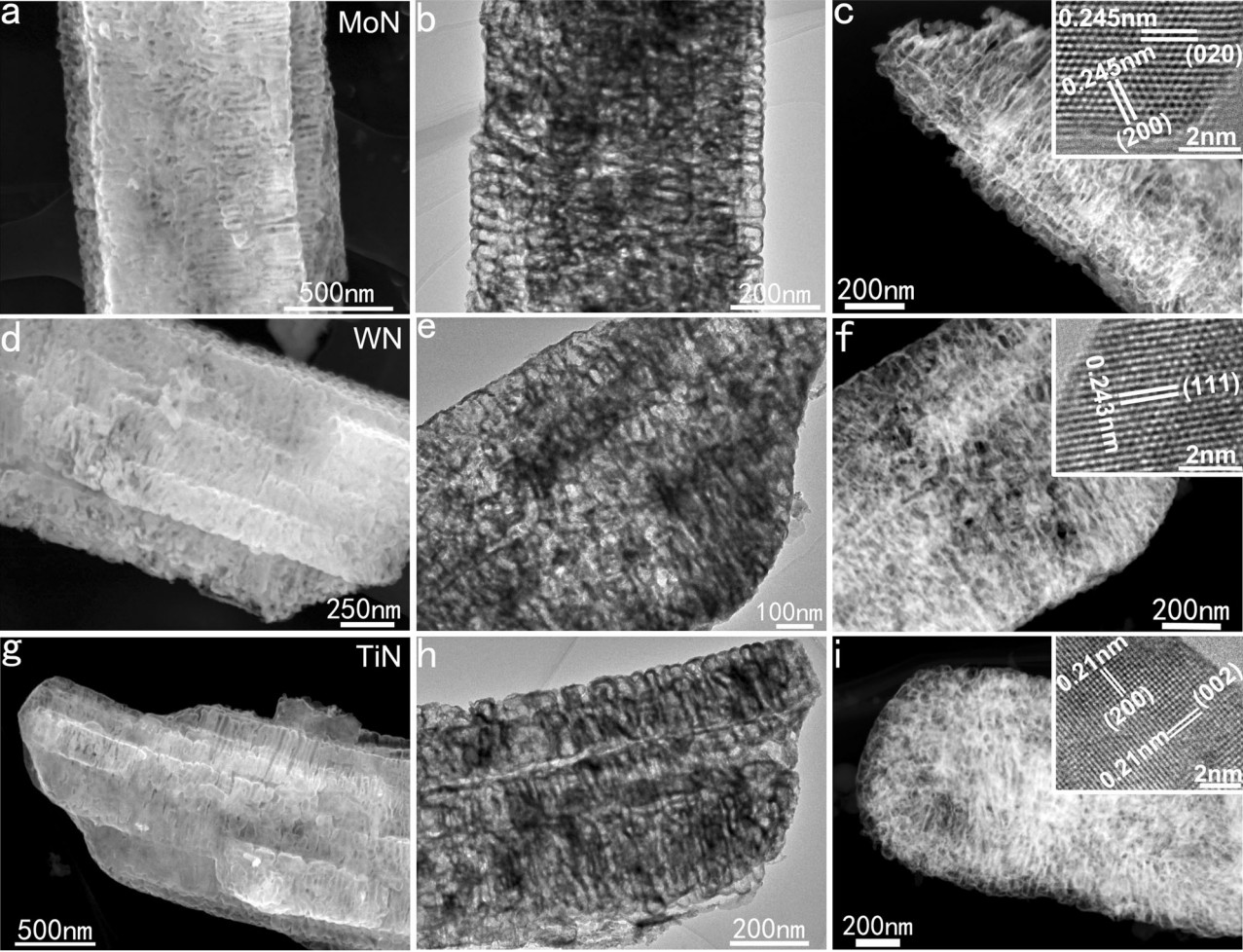

**Fig. 2 Morphology and microstructure of the 3D porous MoN, WN, and TiN. a–c** SEM (**a**), TEM (**b**), HAADF (**c**), and HRTEM (inset) images of the 3D porous MoN. **d–f** SEM (**d**), TEM (**e**), HAADF (**f**), and HRTEM (inset) images of the 3D porous WN. **g–i** SEM (**g**), TEM (**h**), HAADF(**i**), and HRTEM (inset) images of the 3D porous TiN.

formed during the reaction. Under the protection of nitrogen, when the reaction mixture is heated at 290 °C for 0.5 h, the XRD pattern revealed that the obtained product after washing with water and ethanol was ZnO crystals (Fig. 3a), and their formation can be rationally attributed to the hydrolysis of $ZnCl_2 \cdot 6H_2O$ in the molten-salt system. SEM and TEM images show that the ZnO particles exhibit a 3D porous structure (Fig. 3b, c and Supplementary Fig. 10). Although no crystallized VN was detected by XRD at this step, EDS mapping shows that V and N elements evenly distributed in the ZnO porous structures (Fig. 3d). These results demonstrate that micron-scale 3D porous structures supported by the ZnO skeleton and adsorbing a large amount of $N^{3-}$ ions and $V^{3+}$ ions are formed in the first stage of the reaction. When the reaction time was extended to 3 h at 290 °C, Fig. 3e, f reveals that the morphology of the obtained product still maintains a morphology that is basically consistent with the ZnO skeleton, while the XRD pattern shows that the product is a mixture of cubic VN and hexagonal ZnO (Fig. 3g). These results demonstrated that in the second stage of the molten-salt reaction, crystalline VN is formed on the ZnO skeleton. The pure 3D porous VN can be obtained by simply dissolving the ZnO skeletons with dilute HCl.

It should be emphasized that the addition of a small amount of $ZnCl_2 \cdot 6H_2O$ in the anhydrous $ZnCl_2$ is essential to the formation of the 3D porous ZnO structures in the molten-salt system. Controlled experiments confirmed that $ZnCl_2 \cdot 6H_2O$

can also hydrolyze into ZnO 3D porous structures in the $ZnCl_2$ molten-salt even if in the absence of $Li_3N$ and $VCl_3$ (Fig. 3h, i). By contrast, if there is only the presence of anhydrous $ZnCl_2$ as the molten-salt phase, no 3D porous ZnO skeletons formed in the molten-salt system. Correspondingly, since there is no ZnO skeleton as a template, 3D porous VN cannot be formed, and only some micron-level VN particles are obtained (Supplementary Fig. 11). Furthermore, the controlled experiments show that when the mixture of $ZnCl_2 \cdot 6H_2O$ and $ZnCl_2$ was heated in a $N_2$ atmosphere at 290 °C, a large number of tiny pores are generated in the molten-salt (Supplementary Fig. 12a). For comparison, pure $ZnCl_2$ molten-salt did not form pores after heating (Supplementary Fig. 12b). These results suggest that there are gases ($H_2O$ and HCl) generated during the decomposition of $ZnCl_2 \cdot 6H_2O$. These gases may be a key factor in the formation of these TMN porous structures. In addition, the amount of $ZnCl_2 \cdot 6H_2O$ has a significant impact on the final structure of the product. Taking the synthesis of VN as an example, the other parameters remain unchanged. When the amount of $ZnCl_2 \cdot 6H_2O$ is reduced from 2 to 0.5 g, the product has a large amount of solid VN particles in addition to the 3D porous structures (Supplementary Fig. 13a), which may be caused by insufficient ZnO template. On the contrary, when the amount of $ZnCl_2 \cdot 6H_2O$ is increased to 6 g, the product is a honeycomb-like 3D porous vanadium nitride (Supplementary Fig. 13b), which structure should be caused by the

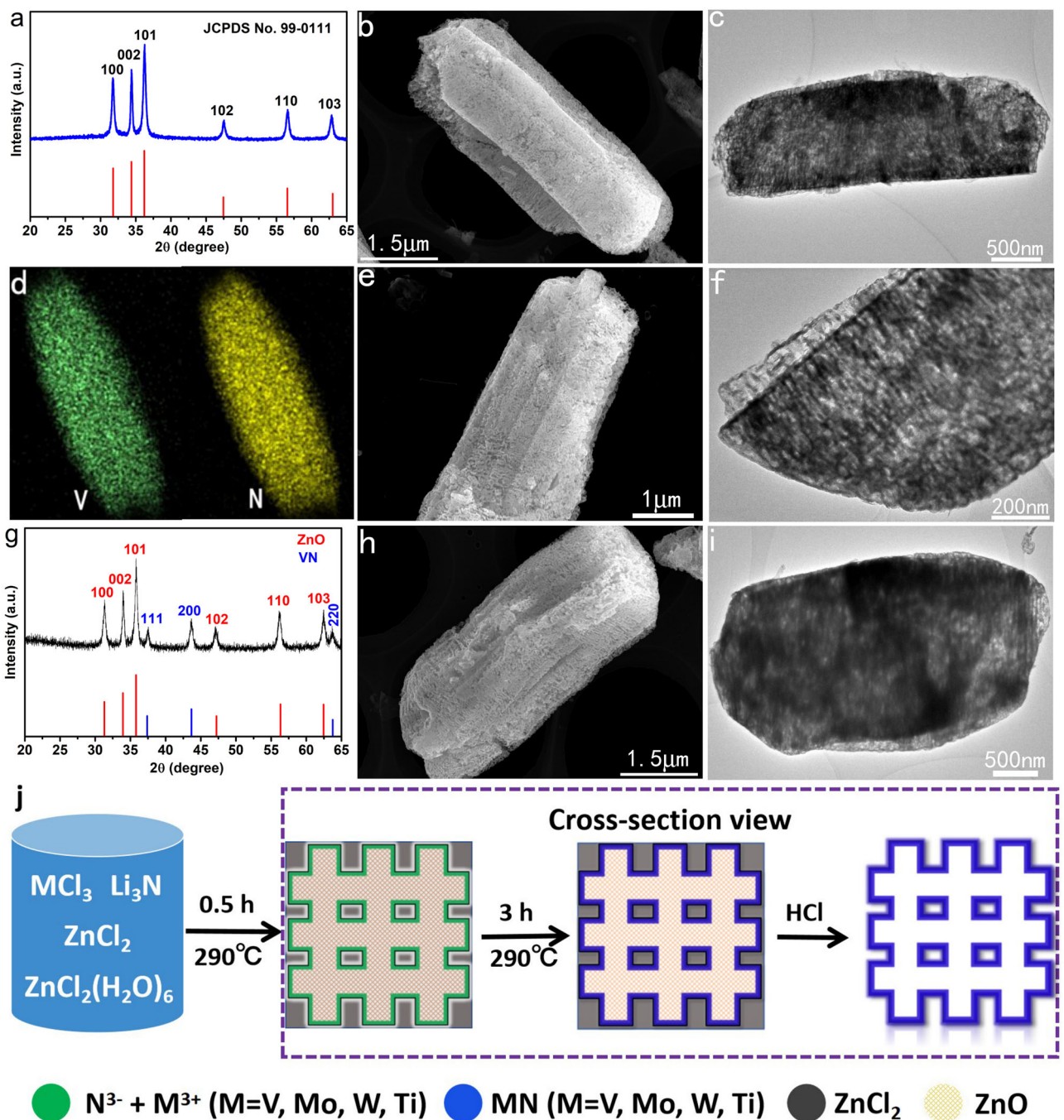

**Fig. 3 Formation mechanism of the 3D porous VN. a–d** XRD pattern, SEM image, TEM image, and EDS mapping of the products formed after 0.5 h reaction. **e–g** SEM image, TEM image, and XRD pattern of the products formed after 3 h reaction. **h, i** SEM and TEM images of the 3D porous ZnO formed in the absence of Li₃N and VCl₃. **j** Formation mechanism scheme of the 3D porous TMNs.

interconnection of ZnO produced by the decomposition of excessive $ZnCl_2 \cdot 6H_2O$.

It can be concluded that the anhydrous $ZnCl_2$ plays the role of molten-salt in the synthesis system, while the $ZnCl_2 \cdot 6H_2O$ plays the role of forming the 3D porous ZnO template. In the traditional process of growing crystalline materials in the molten-salt phase, the presence of trace amounts of water is often regarded as a disadvantage and must be removed. Interestingly, in the current process of growing 3D porous TMNs, trace water has become a key factor, which provides new insights for the in-depth understanding and application of molten-salt reaction to prepare materials with desired structures. We believe that this novel

molten-salt system composed of anhydrous and hydrated salts can provide an effective method for preparing special structural materials at low temperatures. The proposed formation mechanism is summarized in Fig. 3j.

**High surface area and hydrophobic surface of the TMNs.** This micron-scale 3D porous structure avoids the aggregation between nanoparticles, so the obtained TMNs exhibit considerable specific surface area and porosity. Nitrogen adsorption and desorption experiments revealed that the specific surface area and pore volume of the 3D porous VN samples are as high as $156.8 \text{ m}^2 \text{ g}^{-1}$ and $0.81 \text{ cm}^3 \text{ g}^{-1}$ (Fig. 4a). Their pore size distribution range is

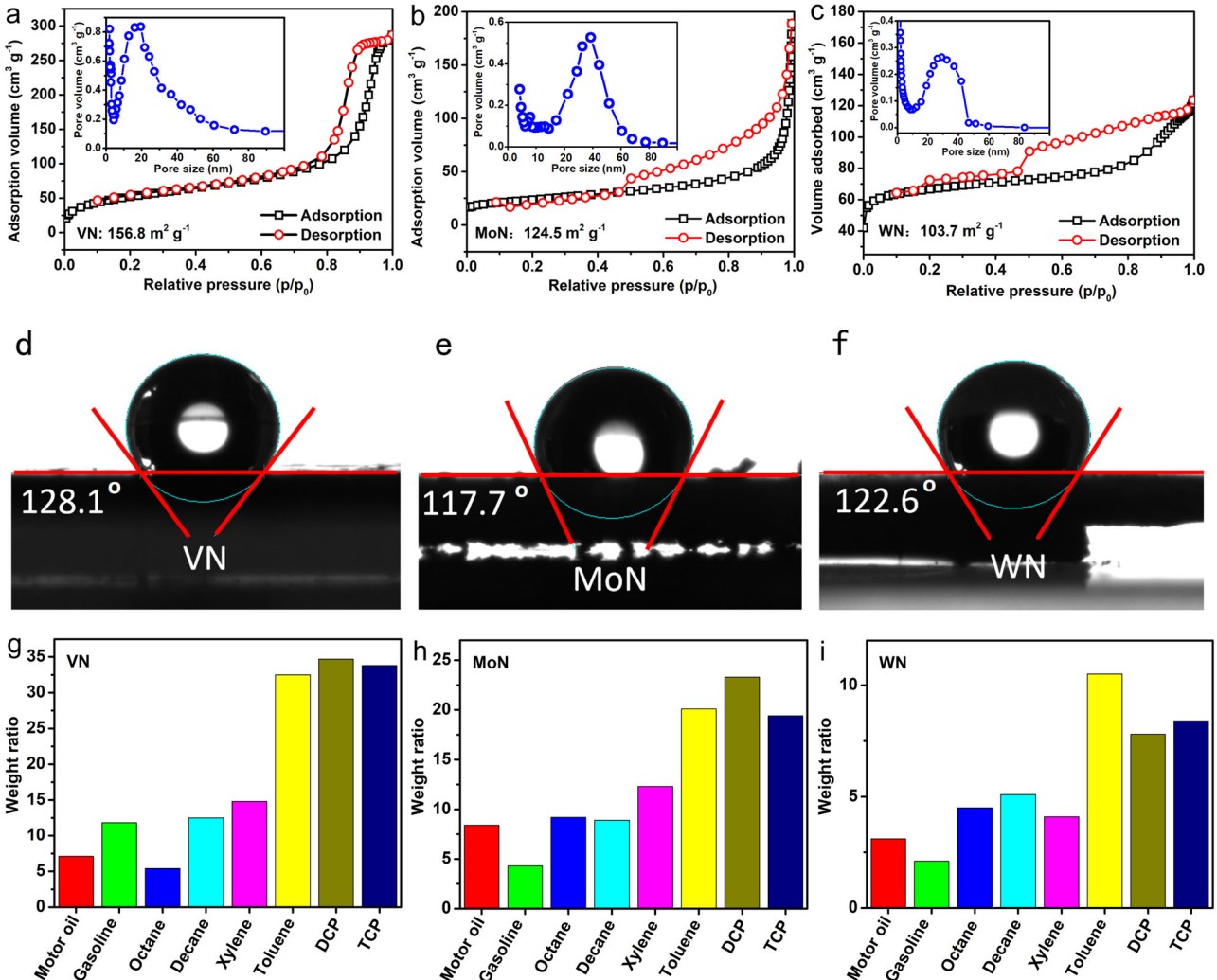

**Fig. 4 Surface area, hydrophobicity, and adsorbability of the 3D porous TMNs. a–c** N$_2$ adsorption and desorption isotherms of the 3D porous VN, MoN, and WN. **d–f** Surface hydrophobicity of the 3D porous VN, MoN, and WN. **g–i** Excellent adsorbability of the 3D porous VN, MoN, and WN.

5–75 nm (inset in Fig. 4a), including mesopores and macropores, which is very conducive to the adsorption and diffusion of substances. Similarly, the 3D porous MoN, WN, and TiN synthesized by this method also show considerable large specific surface area and pore volume, which are 124.5 m$^2$ g$^{-1}$ (0.52 cm$^3$ g$^{-1}$), 103.7 m$^2$ g$^{-1}$ (0.26 cm$^3$ g$^{-1}$), and 135.2 m$^2$ g$^{-1}$ (0.89 cm$^3$ g$^{-1}$) respectively (Fig. 4b, c and Supplementary Fig. 14). Such high surface areas are about three to five times those of the recently reported TMNs prepared with biological templates[19]. Compared with MXenes of TMNs[25–30], these 3D porous TMNs have the advantages of not easy to agglomerate, easy to prepare, and easy to store.

Interestingly, these 3D porous structured TMNs display obvious hydrophobic properties. The contact angle measurement experiments show that the contact angle of the water droplets with the 3D porous VN film is 128.1° (Fig. 4d). Contact angle measurements also confirmed that the surfaces of 3D porous MoN, WN, and TiN also have obvious hydrophobic properties, and the contact angles are 117.7°, 122.6°, and 107.8°, respectively (Fig. 4e, f and Supplementary Fig. 15). Comparative experiments show that none of the commercial TMNs (marked them with C-VN, C-MoN, and C-WN) submicron particles show hydrophobic properties (Supplementary Fig. 16), and they are all water-wettable, which means that the interesting 3D porous structure composed of nanotubes and nanovesicles is a key factor in

achieving this hydrophobic surface. Although the exact mechanism is not yet clear, there is reason to believe that this hydrophobicity has a great relationship with the 3D porous nanostructured surfaces.

Benefiting from the hydrophobic surface, huge specific surface area, and pore volume of the 3D porous VN structures, they show promising application prospects in the adsorption of oil-soluble organic compounds. Figure 4g shows the adsorption capacity of the 3D porous VN microparticles for a variety of oil-soluble organic compounds, which shows that the adsorption performance of the 3D porous VN is amazing. For common oils, such as engine oil and gasoline, the adsorption amount of the VN microparticles is equivalent to 7–12 times their own weight. For a variety of alkanes, such as cyclohexane, octane, sunflower, the 3D porous VN also shows strong adsorption. It is impressive that the 3D porous VN shows extremely high adsorption capacity for high-risk persistent environmental pollutants such as xylene, benzene, dichlorophenol, and trichlorophenol, especially for polychlorophenol, the adsorption amount even reaches 30–35 times of its own weight. Adsorption experiments also confirm that the 3D porous MoN, WN, and TiN possess superior adsorption properties (Fig. 4h, i and Supplementary Fig. 17). This strong adsorption of organics indicates that the 3D porous TMNs have broad application prospects in environmental repair and detection of hazardous substances.

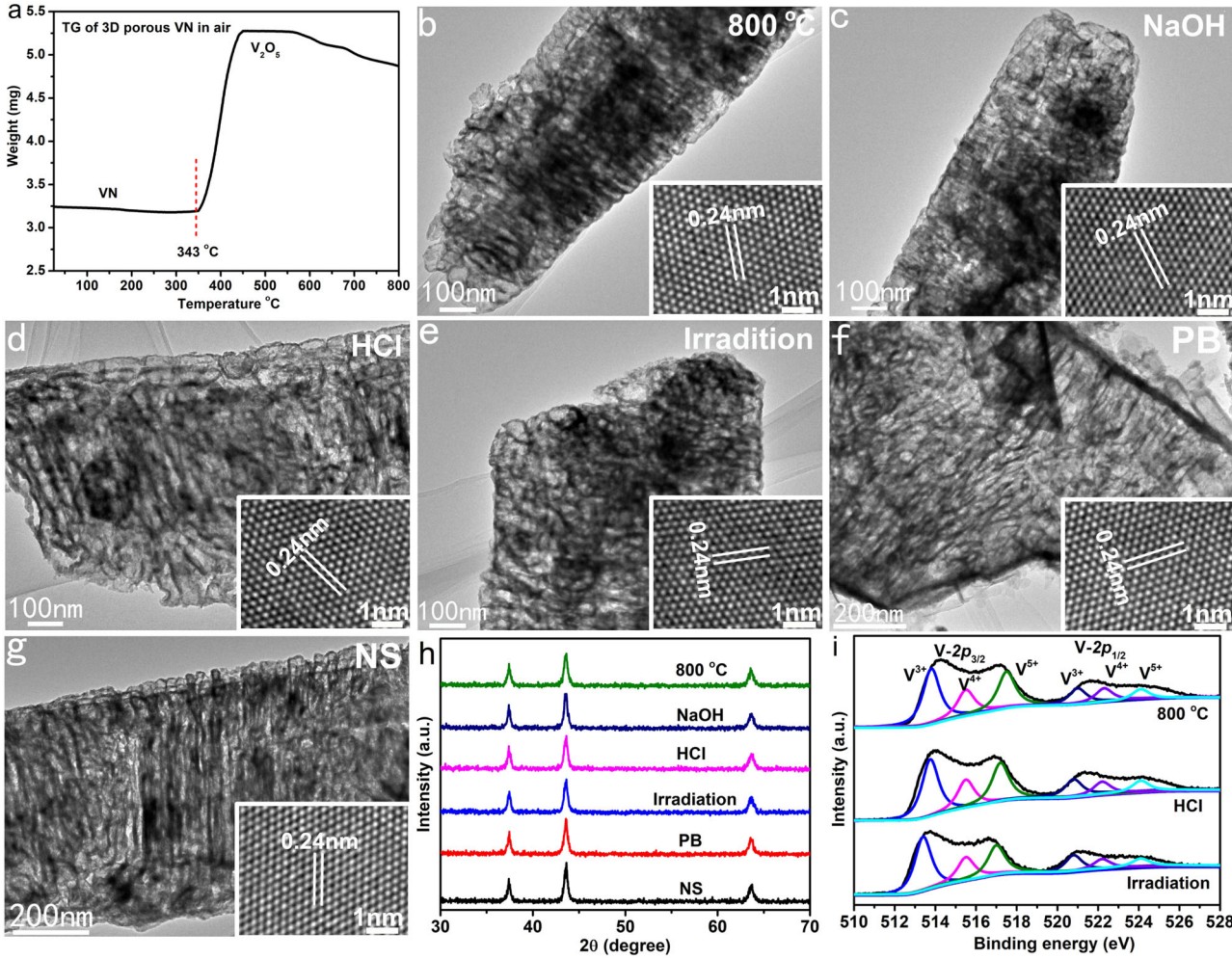

**Fig. 5 Thermal, chemical, and light stability of the 3D porous VN. a** TG curve. **b–g** TEM and HRTEM (inset) images recorded from the 3D porous VN treated at different conditions: **b** 800 °C for 5 h; **c** 6 M NaOH for 5 h; **d** 6 M HCl for 5 h; **e** 20 mW laser (532 nm) for 5 h; **f** 0.3 M PB for 8 h; **g** 0.9% NS for 8 h. **h** XRD patterns of the samples shown in (**b–g**). **i** XPS spectra of the 3D porous VN treated at different conditions.

**Ultrahigh stability of the 3D porous VN**. From the perspective of practical applications, the stability of the material is an important factor. Due to their high lattice energy, bulk TMNs have excellent thermal, mechanical, and chemical stability. In general, when the size of crystals is reduced to the nm-level, their stability will be reduced to a certain extent. Fortunately, the 3D porous VN microparticles possess high thermal, structural, and chemical stability. Thermogravimetric (TGA) experiments show that the 3D porous VN particles have no obvious mass change below 272 °C. Only when the temperature continues to increase to 393 °C, they are gradually oxidized by air to $V_2O_5$ (Fig. 5a). The results of TGA demonstrate that the 3D porous VN sample has good oxidation resistance. When the VN samples were heated at 800 °C in $N_2$ for 5 h, the TEM and HRTEM images proved that their microstructures had not changed appreciably (Fig. 5b), indicating its high thermal stability. The 3D porous VN microparticles can also withstand the corrosion of strong acids and bases. When immersed in 6 M HCl or 6 M NaOH aqueous solution for 5 h, they still maintain a good 3D porous structure and high crystallinity (Fig. 5c, d). When conducting imaging experiments, the SERS substrate material needs to be exposed to laser irradiation for a long time, so the light stability of the material is crucial. Our experimental results show that even when irradiated with 20 mW (normal laser power used in Raman tests <2 mW) laser for 5 h, the VN samples did not undergo structural

damage (Fig. 5e). In addition, the experimental results confirmed that the structure of the VN particles did not change after being placed in common biochemical reagents such as phosphate buffer solution (0.3 M PBS) and normal saline (0.9% NS) for 8 h (Fig. 5f, g). Furthermore, the XRD, XPS, and TGA results also demonstrate the high stability of the 3D porous VN in terms of crystal phase and oxidation state (Fig. 5h, i and Supplementary Figs. 18 and 19). The ultrahigh stability of the 3D porous VN provides the possibility for its practical application.

**Localized-SPR and photothermal conversion effects of the 3D porous VN**. The density functional theory (DFT) calculation results show that there is no band gap in the electronic structure of cubic-phase VN, and the electron density near its Fermi level is very high (Fig. 6a), so it is a typical metallic TMN. It can be seen that most of the density of states near the Fermi level is mainly composed of V-3d orbital electrons, so it can be predicted that VN has high conductivity. The electronic local function (EIF) simulation results also show that the electron gas density around the V atom is high (Fig. 6b), and a large number of V–V metal bonds are formed (indicated by arrows in Fig. 6b). The current–voltage was measured to determine the conductivity of the 3D porous VN. Highly consistent with theoretical prediction, the experimental results confirm that the VN 3D porous

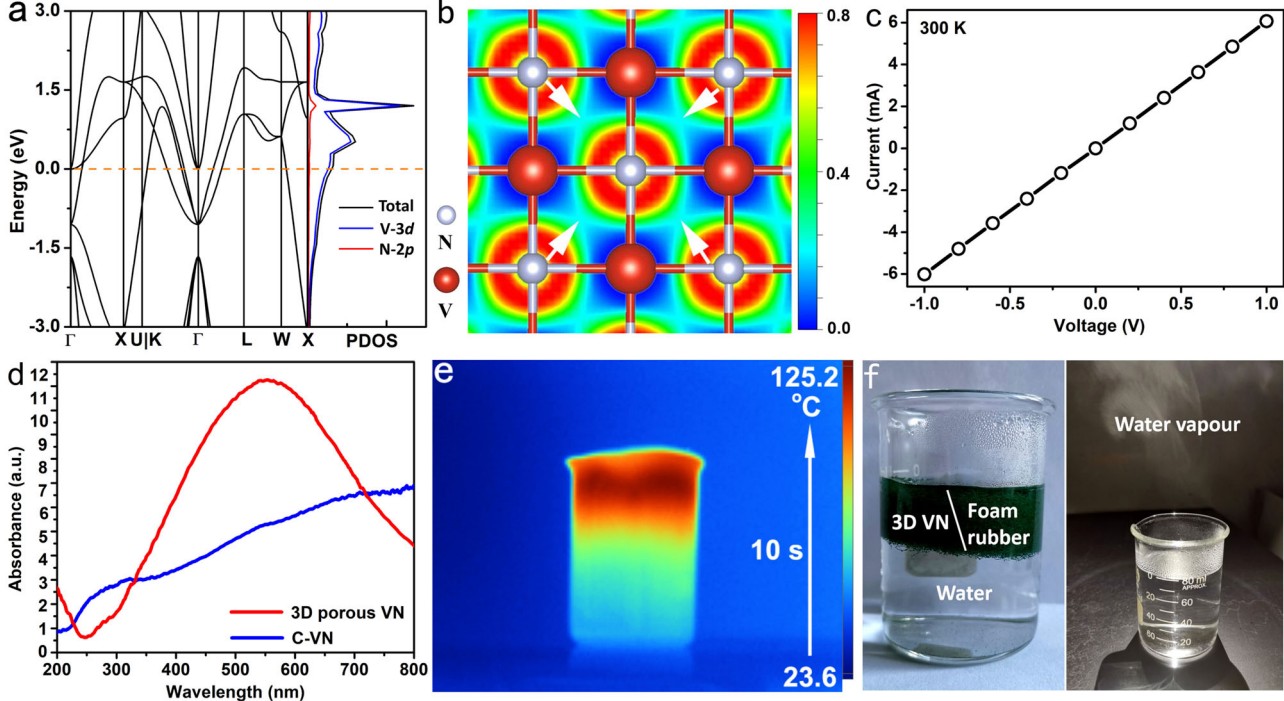

**Fig. 6 SPR and photothermal effects of the 3D porous VN. a**, **b** DOS and EIF of VN. **c** I–V curves of the 3D porous VN at 300 K. **d** UV–Vis absorption of the 3D porous VN. **e** Photothermal effect of the 3D porous VN. **f** Photothermal experimental device (left); no VN particles remain after removing the foam rubber (right).

structures possess excellent conductivity at 300 K, and the temperature dependence of conductivity is linear at the measured temperature, which is highly consistent with metals (Fig. 6c).

Due to the existence of high-density free electrons and a large number of nanocavities, the metallic VN 3D porous structures exhibit a strong localized-SPR effect in the visible light region, with a peak at 553.2 nm (Fig. 6d). In sharp contrast, although commercial VN solid particles also show absorption over a wide range of wavelengths, their intensity is obviously weak and the absorption bands are significantly red-shifted. Although the exact mechanism at this stage is unclear, we believe that the strong localized-SPR effect of the metallic VN in the visible region can be attributed to its unique 3D porous structural features. Under light irradiation, due to the existence of high-density nanogaps or nanopores, a large number of high-intensity electromagnetic field regions will be generated in these positions, that is, formation of hot spots of the electromagnetic field. Therefore, the 3D porous VN microparticles show much stronger SPR effect than solid particles. To our knowledge, this may be the first time that a strong localized-SPR in the visible region of metallic VN has been observed.

Benefiting from this strong localized-SPR effect, the 3D porous VN exhibits superior light-to-heat conversion performance. Under the irradiation of simulated sunlight (xeon light, 2 KW m$^{-2}$), experimental results show that 0.05 g of VN sample can rapidly raise the temperature of the water around it from 23.6 to 125.2 °C in 10 s (Fig. 6e). The photothermal conversion efficiency of the 3D porous VN was determined by the literature method (Supplementary Methods)[31], and the calculated conversion efficiency of the 3D porous VN irradiated by 532 nm laser is 67.3% (Supplementary Fig. 20), which is outstanding even compared with the metal/semiconductor hybrid nanomaterials[32]. Compared with nanoscale photothermal materials, these micron-scale 3D porous VN particles have an obvious advantage as a photothermal agent: due to their larger volume, they are easy to collect and reuse from water. These 3D porous VN powders can be directly dispersed in the common

foam rubber for photothermal experiments (Fig. 6f). When the reaction is over, the foam rubber can be directly removed, and there is almost no residue of VN particles.

**Enhanced Raman scattering effect of the 3D porous VN**. Due to the strong localized-SPR effect, huge specific surface, area and large amount of nanogaps, it can reasonably predict that such 3D porous VN microparticles may possess strong SERS effect[33–37]. To verify this conjecture, we systematically investigated their SERS activity. Rhodamine 6G (R6G), a most common Raman probe molecule, was used to estimate the properties of the 3D porous VN. These VN particles are covered on a common glass slide as a SERS substrate. If there is no special instructions, the excitation wavelength is 532 nm in all Raman experiments, and the power is 0.5 mW. Figure 7a shows a 2D Raman mapping image recorded from the edge of a R6G/VN layer (inset in Fig. 7a). The obtained Raman mapping is obviously divided into two zones: the area covered with 3D porous VN particles shows strong Raman signals of R6G (red zone), whereas the R6G directly placed on the glass did not show any signals (black zone), demonstrating the strong SERS enhancement originated from the 3D porous VN layer. Figure 7b gives the 3D porous VN-based SERS spectrum of 10$^{-7}$ M R6G. A series of Raman scattering peaks are accurately detected, and all the scattering peaks are highly consistent with the Raman spectrum of the certified reference material of R6G (Supplementary Fig. 21). The most recognizable four Raman characteristic peaks of R6G, R$_1$ at 612 cm$^{-1}$, R$_2$ at 772 cm$^{-1}$, R$_3$ at 1362 cm$^{-1}$, and R$_4$ at 1651 cm$^{-1}$, can be clearly observed. Among them, R$_1$ and R$_2$ can be referred to the in-plane and out-of-plane bending motions of C and H atoms of the xanthenes skeleton, respectively, while R$_3$ and R$_4$ can be indexed as the C–C stretching vibrations of aromatic nucleus[38]. By calculation, the Raman enhancement factor of the 3D porous VN substrate to R6G molecules is 5.2 × 10$^7$,

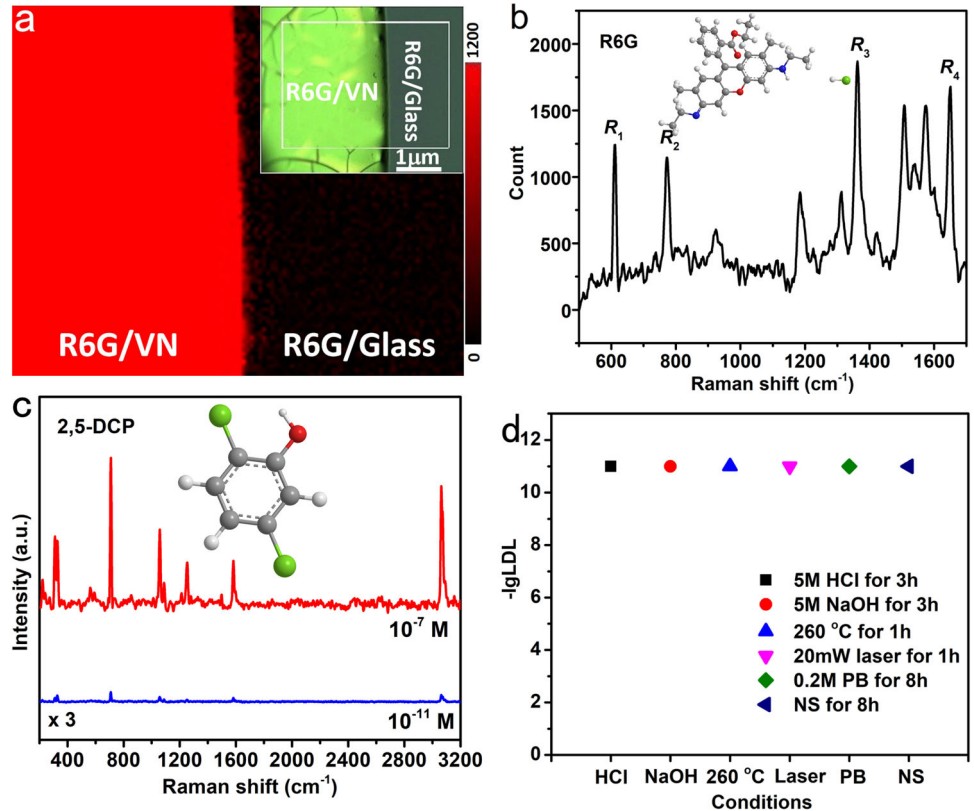

**Fig. 7 Enhanced Raman scattering effect of the 3D porous VN. a** Raman mapping near an edge of a R6G/VN layer, inset: the corresponding optical picture. **b** Raman spectrum of $10^{-8}$ M R6G obtained on the 3D porous VN substrate, inset: molecule structure of R6G. Laser power: 0.5 mW; Integration time: 7 s. **c** Raman spectra of $10^{-7}$ and $10^{-11}$ M 2,5-DCP obtained on the 3D porous VN substrate, inset: molecule structure of 2,5-DCP. Laser power: 0.5 mW; integration time: 5 s for $10^{-7}$ M and 60 s for $10^{-11}$ M. **d** High stability of VN substrate.

which is outstanding among the non-noble-metal SERS substrates (Supplementary Table 1).

Subsequently, we selected the highly risky substance polychlorophenol (2,5-DCP) as the probe molecule. As a hydrophobic SERS substrate, the experimental results show that the 3D porous VN has an ultrasensitive response to 2,5-DCP, with an lowest detection limit (LDL) of $10^{-11}$ M (Fig. 7c). As far as we know, such a high sensitivity is much better than most semiconductor SERS substrates[39–42], and it is also outstanding even compared with noble-metal gold and silver substrates[43–45]. It should be pointed out that the hydrophobic nature of the polychlorophenols makes their SERS signals on traditional hydrophilic noble-metal substrates not obvious. The adsorption experiments mentioned-above have proven that these 3D porous VN microparticles have an extremely high adsorption capacity for 2,5-DCP, so we have reasons to believe that the ultrahigh detection sensitivity can be attributed to the combined effect of high absorption, strong localized-SPR, large specific surface area, and high-density hot spot distribution. The preparation of this super-hydrophobic SERS substrate provides an effective tool for the detection of Raman spectroscopy of lipophilic organic compounds. Furthermore, Raman experiments have proven that the 3D porous VN still maintain $10^{-11}$ M LDL of 2,5-DCP even after harsh processes such as acid-base corrosion, high temperature, laser irradiation, and biochemical reagent immersion (Fig. 7d and Supplementary Fig. 22).

## Discussion

In summary, a simple and universal molten-salt method is established to synthesize high surface area TMNs (VN, MoN,

WN, and TiN) with high crystallinity. It is interesting that all of them show a hydrophobic surface and can efficiently adsorb organic compounds. As a new function, the 3D porous VN SERS substrate shows unexpectedly high sensitivity and high stability for polychlorophenol. It should be pointed out that this $ZnCl_2$ molten-salt system greatly reduces the melting temperature compared to the conventional NaCl system, and the addition of a small amount of $ZnCl_2 \cdot 6H_2O$ containing crystal water provides an easy-to-form and remove template for the synthesis of 3D porous TMNs. This novel synthesis method and the obtained interesting structure provide a valuable reference for exploring the properties and potential applications of 3D porous TMNs.

## Methods

**Synthesis of 3D porous TMNs.** In a typical synthesis, 1 mmol of $VCl_3$ and 1 mmol of $Li_3N$ are dispersed in a mixture of 40 g of anhydrous $ZnCl_2$ and 2 g of $ZnCl_2 \cdot 6H_2O$. In a glove box protected by $N_2$, the precursor mixture was sufficiently ground and placed in a 60-mL corundum crucible. After being sealed, it was placed in an electronic temperature control furnace for heating. The heating rate was 1 °C/min, and it was held at 180 °C for 1 h. Then, continue heating to 290 °C at a rate of 2 °C min$^{-1}$ and hold for 3 h. The obtained product obtained by the reaction was immersed in the HCl solution (3M) for 1 h, and then washed with distilled water and ethanol three times, and finally the blue–black product was dried in an oven. When synthesizing MoN, WN, and TiN, it is only necessary to replace the vanadium source with a molybdenum source ($MoCl_3$), tungsten source ($WCl_3$), and a titanium source ($TiCl_3$).

**SERS tests.** To study the SERS properties of these as-synthesized 3D porous VN samples, a confocal micro Raman spectrometer (Renishaw-inVia Reflex) is used as the measuring instrument. In all SERS tests, unless specifically stated, the excitation wavelength is 532 nm, laser power is 0.5 mW, and the specification of the objective is ×100 L. A series of standard solution (aqueous) of R6G with concentrations of $10^{-7}$–$10^{-11}$ M were used as the probe molecules. To improve the signal

reproducibility and uniformity, the glass sheet covered with VN porous samples (50 mg) was immersed into a probe solution (40 mL) to be measured for 10 min, then taken out and dried in air for 5 min. In all SERS tests, the laser beam is perpendicular to the top of the sample to be tested with a resultant beam spot diameter of 5 μm. The fluorescent background of the probe molecule is deducted by the software that comes with the instrument.

**Material characterization**. These samples were measured by a variety of characterization techniques. XRD patterns of the products were obtained on a Bruker D8 focus X-ray diffractometer by using CuKa radiation (l = 1.54178 Å). SEM images and EDS were obtained on a Hitachi S-4800. TEM and HRTEM characterizations were performed with a Tecnai G F30 operated at 300 kV. Ultraviolet–Vis absorption spectra were recorded with a Shimadzu UV3600. XPS experiments were performed in a ESCALab250Xi using monochromated Al Kα X-rays at hυ = 1486.6 eV. Peak positions were internally referenced to the C1s peak at 284.6 eV. The specific surface area was measured in a Micro Tristar II 3020. Raman spectra were recorded from Renishaw-inVia Qontor. High-resolution mass spectrometry was obtained from a Thermo-Q Exactive Focus (NanoESI, m/z 50-500).

**Electronic structure calculations**. The details of all DFT calculations and EIF simulation are provided in Supporting Information.

**Photothermal conversion efficiency**. The calculation details of photothermal conversion efficiency of the 3D porous VN are provided in Supporting Information.

## Data availability
The data that support the findings of this study are available from the corresponding author upon request. All reported data are included in the paper. The source data can be downloaded from: https://pan.baidu.com/s/1F_ElycA9roFRpBi_CbFDIg (Code: dzvu).

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

## Acknowledgements

This work received financial support from the National Key Research and Development Program of China (2017YFF0210003) and the Science Foundation of Chinese Academy of Inspection and Quarantine (2019JK004).

## Author contributions

G.X. proposed and designed the project. H.G. and J.H. prepared materials. W.L. and Q.K. performed ultraviolet–vis, XRS, XRD, SEM, and TEM characterization. H.B. conducted SERS and EFs measurement. W.Y. performed electronic structure calculations. All authors critically evaluated the manuscript.

## Competing interests

The authors declare no competing interests.
