## [Peer Review File · Nature Communications]

REVIEWER COMMENTS

Reviewer #1 (Remarks to the Author):

The authors have developed a new method for the synthesis of TMN materials, which have great large specific surface area and pore volume. As we know, high surface area and pore volume will bring super extensive applications to materials in various fields, such catalysis, sensing, gas and energy storage. This method seems novel and easy to handle, in addition, the structure of the as obtained TMN and the formation mechanism the author inferred are also very interesting. I think this work may provide a valuable reference for other transition metal materials of special structure, and should be accepted after a minor revision:

1. Why $\text{ZnCl}_2 \cdot 6\text{H}_2\text{O}$ plays a key role in forming the 3D porous ZnO? Is it because the decomposition products of $\text{ZnCl}_2 \cdot 6\text{H}_2\text{O}$ is different from that of ZnCl_2 ?
2. The XRD pattern of the product (before acid etching) with absent $\text{ZnCl}_2 \cdot 6\text{H}_2\text{O}$ is suggested.

Reviewer #2 (Remarks to the Author):

This work reported a general and mild molten-salt route to prepare 3D porous VN, MoN, WN and TiN with high surface area and porosity at quite low temperature of 290 °C. The 3D porous TMNs showed some interesting properties, including superhydrophobic surface with high adsorption amount of organic compounds, strong surface plasmon resonance, enhanced Raman scattering effect and good photothermal conversion efficiency. Therefore, the publication of this manuscript can be considered after revision.

1. This work used Li_3N and anhydrous transition metal chlorides as the starting materials. These compounds are not of wide availability, which undermined the merit of this work. The reaction was simply a metathesis reaction which can be carried out in many different conditions, such as direct solid state reaction. Moreover, solid-gas reaction using metal chlorides and NH_3 is also possible to obtain transition metal nitrides. What is the advantage of using molten salt. Moreover, the low-temperature chlorides molten salts were also reported in other inorganic synthesis (e.g. Nature 2018, 556, 355; J. Mater. Chem. A, 2020, 8, 6597–6606; Angew. Chem. Int. Ed. 2020, 59, 1975-1979; Energy Environ. Sci. 2015, 8, 3187-3191). Thus, the authors should justify the novelty and advantage of the new synthetic method, particularly the advantage of using molten salt reaction.
2. It's very interesting that 3D porous structure form during the synthesis process. The author claimed their formation could be attributed to the hydrolysis of $\text{ZnCl}_2 \cdot 6\text{H}_2\text{O}$ to generate ZnO porous skeleton. But more detailed reasons should be discussed. The hydrolysis of $\text{ZnCl}_2 \cdot 6\text{H}_2\text{O}$ probably releases HCl and H_2O vapour. Whether is the formation of porous structure related to gas release?
3. Since the in-situ formed 3D porous ZnO played the role of template, the using amount of $\text{ZnCl}_2 \cdot 6\text{H}_2\text{O}$ was important to control the porous structure and its surface area. Why did the author choose the current ratio (2 g $\text{ZnCl}_2 \cdot 6\text{H}_2\text{O}$, 40 g anhydrous ZnCl_2 , 1 mmol VCl_3 and 1 mmol Li_3N)? The influence of $\text{ZnCl}_2 \cdot 6\text{H}_2\text{O}$ amount should be investigated. In addition, ZnCl_2 was used in a very large quantity. Is it possible to reduce the using amount for the cost consideration?
4. In order to illustrate the advantage of 3D porous VN SERS substrate, authors could provide a systematic comparison (e.g. Table) of their work vs. earlier publications.
5. The second paragraph of Page 18 is the conclusion of this work, but the subheading was written as "Discussion". Please revise it.

Reviewer #3 (Remarks to the Author):

General comments: The manuscript describes a new method for the synthesis of porous metal nitrides using ZnCl_2 as a template. The authors attribute superhydrophobic properties, adsorption of organic

compounds, photo thermal conversion and SERS activity to the transition metal nitrides synthesised by this route. The emphasis is on the new route of synthesis, however, purity and phase confirmation of synthesized metal nitrides is not proven unambiguously. Experimental details are lacking, particularly on the proposed applications.

Recommendation: Reconsider after major revision

Specific Comments:

1. Given that several applications are included in the manuscript, introduction can be a bit more elaborate including the adsorption capacities, photo thermal conversion and SERS activity of reported metal nitrides.
2. Atomic ratio of VN (and it is calculated only for VN, why not for other nitrides also) is calculated from EDS spectrum (Figure 1l), but in EDS spectrum there is an overlap of V L alpha(0.51 keV) with O K alpha (0.52 keV), hence elemental composition shall be further confirmed with some other technique like XPS.
3. Though the phase of the material is confirmed by XRD, presence of oxygen cannot be ruled out, particularly when ZnO is formed as intermediate during the synthesis, which is further removed by washing with 3M HCl. How will you explain the V⁴⁺ and V⁵⁺ ions detected in the XPS spectrum of VN (which is not a small amount as claimed by the authors)? XPS spectrum of other metal nitrides (Fig.S8) also shows the presence of metal oxides. Estimate elemental composition (of all elements) using XPS
4. How the complete removal of ZnO is ensured during the washing step?
5. The proposed formation mechanism (Fig.3j) is confusing. The picture gives the impression of ZnO particles in MN framework.
6. Attributing superhydrophobic properties to the TMNs is not correct as all the synthesised metal nitrides have contact angle in the range of 107° to 128°. A material is said to be superhydrophobic, when the contact angle is more than 150°.
7. Experimental details of adsorbability, photothermal effect and hydrophobicity studies?
8. In Fig.S14 caption and label in the figure do not match
9. The stability of VN to harsh environment, heat and corrosion is mainly characterized by V2p XPS spectrum (Figure 5i). It is better to include XPS survey spectra and corresponding N1s and O1s spectra also to support the claim.
10. The authors claim 'high' thermal stability for the TMNs. The experimental data provided shows thermal stability only upto about 300°C. Include a TGA under inert atmosphere to support the claim of stability upto 800°C.
11. The weight increase as shown in the TG in air for VN (Fig.5a), is not matching with theoretical value (this inference is only based on the figure provided, quantitative data is not available). This supports my earlier observation that the VN phase is not pure. Please do a quantitative estimate of weight increase due to oxidation and compare it with theoretical value
12. To measure SERS activity, enhancement factor of VN for model compound is to be included in the manuscript.
13. The SERS sensitivity of 10⁻¹¹M is questionable. The experiment includes dipping of VN coated glass plate into the sample solution. What is the quantity of VN on glass plate and what is the quantity of the solution used? According to the authors, 'VN microparticles have extremely high adsorption capacity for DCP'. Therefore, the enhancement shown in Raman spectrum could be due to the extremely high concentration of DCP on VN particles. This is also evident from Fig.7a, where all the R6G is detected on VN and not even a trace is detected on the glass surface. Also, the authors report a beam spot size of 5 mm for the Raman measurement, which is unusual.
14. Figure 7d presents LDL (y-axis) vs treatment conditions (x-axis), Raman spectra to be included to show the detection of PCPs and high stability of VN substrate as claimed by the authors
15. IR and Mass spectrometry results are not discussed in manuscript but included in experimental section
16. Out of curiosity, have the authors attempted desorption of adsorbed organic species.
17. Typographic errors to be corrected and grammar check required

We are very grateful for the questions and suggestions raised by the reviewers.

The following is our response to these questions and suggestions one by one.

Reviewer-1's specific comments

The authors have developed a new method for the synthesis of TMN materials, which have great large specific surface area and pore volume. As we know, high surface area and pore volume will bring super extensive applications to materials in various fields, such catalysis, sensing, gas and energy storage. This method seems novel and easy to handle, in addition, the structure of the as obtained TMN and the formation mechanism the author inferred are also very interesting. I think this work may provide a valuable reference for other transition metal materials of special structure, and should be accepted after a minor revision:

1. Why $\text{ZnCl}_2 \cdot 6\text{H}_2\text{O}$ plays a key role in forming the 3D porous ZnO? Is it because the decomposition products of $\text{ZnCl}_2 \cdot 6\text{H}_2\text{O}$ is different from that of ZnCl_2 ?

Reply: We are very grateful to the reviewer for the this question. $\text{ZnCl}_2 \cdot 6\text{H}_2\text{O}$ plays a very important role in the formation of the 3D porous TMNs. As the reviewer said, the decomposition products of $\text{ZnCl}_2 \cdot 6\text{H}_2\text{O}$ and ZnCl_2 are different. Research on the formation mechanism found that due to the presence of crystal water, $\text{ZnCl}_2 \cdot 6\text{H}_2\text{O}$ will be hydrolyzed to form $\text{Zn}(\text{OH})_2$ in the high-temperature molten-salt system, and then further pyrolyzed to form 3D porous ZnO. Using these 3D porous ZnO skeleton as template, TMN/ZnO composite material is formed, and finally the ZnO skeletons are removed by acid washing to obtain 3D porous TMNs. In contrast, when ZnCl_2 does not contain water of crystallization, the ZnCl_2 in the molten salt system cannot

be hydrolyzed to produce $\text{Zn}(\text{OH})_2$, and then the ZnO skeleton cannot be produced as a template, so 3D porous TMNs cannot be finally obtained. Therefore, it is the addition of a small amount of zinc chloride with crystal water in a large amount of anhydrous zinc chloride that is the key to the formation of these 3D porous TMNs. Related discussions have been added to our revised version.

Page 8-10 (yellow part): It should be emphasized that the addition of a small amount of $\text{ZnCl}_2 \cdot 6\text{H}_2\text{O}$ in the anhydrous ZnCl_2 is essential to the formation of the 3D porous ZnO structures in the molten-salt system. Controlled experiments confirmed that $\text{ZnCl}_2 \cdot 6\text{H}_2\text{O}$ also can hydrolyze into ZnO 3D porous structures in the ZnCl_2 molten-salt even if in the absence of Li_3N and VCl_3 (Figure 3h-i). By contrast, If there is only the presence of anhydrous ZnCl_2 as the molten-salt phase, no 3D porous ZnO skeletons formed in the molten-salt system. Correspondingly, since there is no ZnO skeleton as a template, 3D porous VN cannot be formed, and only some micron-level VN particles are obtained (Supplementary Figure 11). It can be concluded that the anhydrous ZnCl_2 plays the role of molten-salt in the synthesis system, while the $\text{ZnCl}_2 \cdot 6\text{H}_2\text{O}$ plays the role of forming the 3D porous ZnO template. In the traditional process of growing crystalline materials in the molten-salt phase, the presence of trace amounts of water is often regarded as a disadvantage and must be removed. Interestingly, in the current process of growing 3D porous TMNs, trace water has become a key factor, which provides new insights for the in-depth understanding and application of molten salt reaction to prepare materials with desired structures.

2. The XRD pattern of the product (before acid etching) with absent $\text{ZnCl}_2 \cdot 6\text{H}_2\text{O}$ is

suggested.

Reply: We are very grateful to the reviewer for this suggestion. According to this suggestion, the XRD pattern of the product (before acid etching) with absent $\text{ZnCl}_2 \cdot 6\text{H}_2\text{O}$ have been added into the revised Supporting Information (*please see Supplementary Figure 11c in Supporting Information*). The results show that the sample before pickling is a mixture of ZnCl_2 , NaCl , and VN, which shows that anhydrous ZnCl_2 only melts and does not decompose at our experimental temperature (290 °C).

Supplementary Figure

the acid washing, which indicates that it is a mixture of VN, ZnCl_2 , and NaCl .

Reviewer-2's specific comments

This work reported a general and mild molten-salt route to prepare 3D porous VN, MoN, WN and TiN with high surface area and porosity at quite low temperature of 290 °C. The 3D porous TMNs showed some interesting properties, including superhydrophobic surface with high adsorption amount of organic compounds, strong surface plasmon resonance, enhanced Raman scattering effect and good photothermal

conversion efficiency. Therefore, the publication of this manuscript can be considered after revision.

1. This work used Li_3N and anhydrous transition metal chlorides as the starting materials. These compounds are not of wide availability, which undermined the merit of this work. The reaction was simply a metathesis reaction which can be carried out in many different conditions, such as direct solid state reaction. Moreover, solid-gas reaction using metal chlorides and NH_3 is also possible to obtain transition metal nitrides. What is the advantage of using molten salt. Moreover, the low-temperature chlorides molten salts were also reported in other inorganic synthesis (e.g. Nature 2018, 556, 355; J. Mater. Chem. A, 2020,8, 6597–6606; Angew. Chem. Int. Ed. 2020, 59, 1975-1979; Energy Environ. Sci. 2015, 8, 3187-3191). Thus, the authors should justify the novelty and advantage of the new synthetic method, particularly the advantage of using molten salt reaction.

Reply: We are very grateful to the reviewers for these questions, which are very helpful in improving the quality of this paper. We carefully read the literatures provided by the reviewer and found that they were all synthesized by molten salt reaction to synthesize transition-metal chalcogenides, phosphides and silicon nanostructures, which shows that molten salt phase reaction is a very good synthesis route. *The four references have been added into the revised paper.* Indeed, as the reviewer said, both gas-solid and solid-solid reactions can be used to synthesize TMN, but these traditional reactions require harsh conditions, such as high temperature (~ 1000 °C) high pressure (several GPa).^{14,15} With the untiring efforts, researchers have

recently succeeded in preparing metallic TMNs under atmospheric pressure by improved gas-solid reaction routes.¹⁶⁻¹⁹ For example, Zhou and Gogotsi et al. reported salt-templated synthesis of 2D TMN nanostructures.^{17,18} Liu et al. prepared TMNs architectures by using diatomite as template.¹⁹ Although several interesting TMN materials have been prepared by the newly developed methods, the reaction temperature is still as high as 650 °C. Therefore, the development of a more gentle reaction route is very necessary for the practical application of TMNs. Although molten-salt systems have been used to synthesize transition metal sulfides and TMNs, the reaction temperature is often as high as 600 °C due to the higher melting point of NaCl as molten-salt and the higher decomposition temperature of ammonia as the nitrogen source. In the current reaction system, ZnCl₂ with a very low melting point (280-290 °C) is used as molten-salt, while Li₃N and metal chlorides with high reaction activity are used as precursors, which greatly reduces the formation temperature of TMN. Moreover, we added a small amount of ZnCl₂·6H₂O to the anhydrous ZnCl₂. Due to the hydrolysis of ZnCl₂·6H₂O, a 3D porous ZnO skeleton is produced, which provides a template for the formation of 3D porous TMNs. According to the reviewer's suggestions, we justify the novelty and advantage of the new synthetic method in the *Introduction* and *Conclusion* of the article:

Page 4 (yellow part): Molten-salt reaction route has been proved to be a very effective synthesis route,²⁰⁻²³ but there is no low-temperature molten-salt synthesis method for TMNs with highly porous structure.

Page 18 (yellow part): It should be pointed out that this ZnCl₂ molten-salt system

greatly reduces the melting temperature compared to the conventional NaCl system, and the addition of a small amount of $\text{ZnCl}_2 \cdot 6\text{H}_2\text{O}$ containing crystal water provides an easy-to-form and remove template for the synthesis of 3D porous TMNs.

New Added References

(20) Zhou, J. G. et al. A library of atomically thin metal chalcogenides. *Nature* **556**, 355-359 (2018).

(21) Zhou, J.; Zhao, H. Y.; Lin, N.; Li, T. Q.; Li, Y.; Jiang, S.; Tian J. & Qian, Y. T. Silicothermic reduction reaction for fabricating interconnected Si-Ge nanocrystals with fast and stable Li-storage. *J. Mater. Chem. A* **8**, 6597-6606 (2020).

(22) Liu, Z.; Yang, S.; Sun, B.; Yang, P.; Zheng, J. & Li, X. G. Low-temperature synthesis of honeycomb $\text{CuP}_2@\text{C}$ in molten ZnCl_2 salt for high-performance lithium ion batteries. **59**, 1975-1979 (2020).

(23) Lin, N.; Han, Y.; Zhou, J.; Zhang, K.; Xu, T.; Zhu, Y. C. & Qian, Y. T. A low temperature molten salt process for aluminothermic reduction of silicon oxides to crystalline Si for Li-ion batteries. *Energy Environ. Sci.* **8**, 3187-3191 (2015).

2. It's very interesting that 3D porous structure form during the synthesis process. The author claimed their formation could be attributed to the hydrolysis of $\text{ZnCl}_2 \cdot 6\text{H}_2\text{O}$ to generate ZnO porous skeleton. But more detailed reasons should be discussed. The hydrolysis of $\text{ZnCl}_2 \cdot 6\text{H}_2\text{O}$ probably releases HCl and H_2O vapour. Whether is the formation of porous structure related to gas release?

Reply: This is a very good suggestion, and we are very grateful to the reviewers for this suggestion. It should be emphasized that the addition of a small amount of

$\text{ZnCl}_2 \cdot 6\text{H}_2\text{O}$ in the anhydrous ZnCl_2 is essential to the formation of the 3D porous ZnO structures in the molten-salt system. Controlled experiments confirmed that $\text{ZnCl}_2 \cdot 6\text{H}_2\text{O}$ also can hydrolyze into ZnO 3D porous structures in the ZnCl_2 molten-salt even if in the absence of Li_3N and VCl_3 (Figure 3h-i). By contrast, If there is only the presence of anhydrous ZnCl_2 as the molten-salt phase, no 3D porous ZnO skeletons formed in the molten-salt system. Correspondingly, since there is no ZnO skeleton as a template, 3D porous VN cannot be formed, and only some micron-level VN particles are obtained (Supplementary Figure 11). It can be concluded that the anhydrous ZnCl_2 plays the role of molten-salt in the synthesis system, while the $\text{ZnCl}_2 \cdot 6\text{H}_2\text{O}$ plays the role of forming the 3D porous ZnO template. As the reviewer said, we also believe that the pores in the ZnO 3D porous structures are caused by H_2O and HCl gas generated by the decomposition of $\text{ZnCl}_2 \cdot 6\text{H}_2\text{O}$. However, since we are currently unable to perform in-situ observations, we have not yet obtained direct evidence of the existence of water vapor. Moreover, since the reaction temperature far exceeds the boiling point of water, we did not detect the water component in the final product. Nevertheless, we still hope to find some indirect evidence that gas has existed. To this end, additional experiments were carried out. The experimental results show that when the mixture of $\text{ZnCl}_2 \cdot 6\text{H}_2\text{O}$ and ZnCl_2 is heated in a nitrogen atmosphere at 290 °C, a large number of tiny pores are generated in the molten-salt. For comparison, pure ZnCl_2 molten-salt did not form pores after heating. These experimental results indirectly prove that there is gas generated during the formation of TMN. Relative discussion and Figures have been added into the revised paper:

Page 9 (yellow part): Furthermore, the controlled experiments show that when the mixture of $\text{ZnCl}_2 \cdot 6\text{H}_2\text{O}$ and ZnCl_2 was heated in a N_2 atmosphere at $290\text{ }^\circ\text{C}$, a large number of tiny pores are generated in the molten-salt (*Figure S12a*). For comparison, pure ZnCl_2 molten-salt did not form pores after heating (*Figure S12b*). These results suggest that there is gases (H_2O and HCl) generated during the decomposition of $\text{ZnCl}_2 \cdot 6\text{H}_2\text{O}$. These gases may be a key factor in the formation of these TMN porous structures.

Supplementary Figure 12. SEM images of the molten salt after cooling. (a) The precursor of $\text{ZnCl}_2 \cdot 6\text{H}_2\text{O}$ and ZnCl_2 (mass ratio is 1:20) was transformed into the mixture of ZnO and ZnCl_2 during the melting process, and a large number of holes are formed. (b) When the molten salt is only ZnCl_2 , the molten salt is very transparent and smooth after cooling, and no holes are formed.

3. Since the in-situ formed 3D porous ZnO played the role of template, the using amount of $\text{ZnCl}_2 \cdot 6\text{H}_2\text{O}$ was important to control the porous structure and its surface area. Why did the author choose the current ratio (2 g $\text{ZnCl}_2 \cdot 6\text{H}_2\text{O}$, 40 g anhydrous ZnCl_2 , 1 mmol VCl_3 and 1 mmol Li_3N)? The influence of $\text{ZnCl}_2 \cdot 6\text{H}_2\text{O}$ amount should be investigated. In addition, ZnCl_2 was used in a very large quantity. Is it possible to

reduce the using amount for the cost consideration?

Reply: We are very grateful to the reviewers for these questions, which are very helpful in improving the quality of this paper. Indeed, as the reviewer said, the amount of $\text{ZnCl}_2 \cdot 6\text{H}_2\text{O}$ has a great influence on the final product structure. The current ratio (2 g $\text{ZnCl}_2 \cdot 6\text{H}_2\text{O}$, 40 g anhydrous ZnCl_2 , 1 mmol VCl_3 and 1 mmol Li_3N) was chosen because the porous structure of the product formed under this condition is the most obvious. According to the reviewer's suggestion, we characterized the product structure of vanadium nitride obtained under different dosages of $\text{ZnCl}_2 \cdot 6\text{H}_2\text{O}$ by SEM. The characterization results show that when the amount of $\text{ZnCl}_2 \cdot 6\text{H}_2\text{O}$ is only 0.5 g, in addition to the 3D porous vanadium nitride, a large number of solid vanadium nitride particles are produced. This may be caused by insufficient ZnO template. On the contrary, when the amount of $\text{ZnCl}_2 \cdot 6\text{H}_2\text{O}$ is increased to 6 g, the product is a honeycomb-like 3D porous vanadium nitride. This structure should be caused by the interconnection of ZnO produced by the decomposition of excessive zinc chloride. Although these honeycomb vanadium nitrides also have a 3D porous structure, the amount of zinc chloride used in their synthesis process has been greatly increased, increasing production costs. Regarding the amount of anhydrous ZnCl_2 , the amount is indeed relatively large at present. But from an industrial perspective, these ZnCl_2 molten salts can be reused, so the cost can be greatly reduced. The relative discussion has been added into the revised paper:

Page 10 (yellow part): In addition, the amount of $\text{ZnCl}_2 \cdot 6\text{H}_2\text{O}$ has a significant impact on the final structure of the product. Taking the synthesis of VN as an example,

the other parameters remain unchanged. When the amount of $\text{ZnCl}_2 \cdot 6\text{H}_2\text{O}$ is reduced from 2 g to 0.5 g, the product has a large amount of solid VN microparticles (*Supplementary Figure 13a*) in addition to the 3D porous structures, which may be caused by insufficient ZnO template. On the contrary, when the amount of $\text{ZnCl}_2 \cdot 6\text{H}_2\text{O}$ is increased to 6 g, the product is a honeycomb-like 3D porous VN (*Supplementary Figure 13b*), which structure should be caused by the interconnection of ZnO produced by the decomposition of excessive $\text{ZnCl}_2 \cdot 6\text{H}_2\text{O}$.

Supplementary Figure 13. SEM images of VN samples obtained under different $\text{ZnCl}_2 \cdot 6\text{H}_2\text{O}$ contents. (a) 0.5 g. (b) 6g. In these comparative experiments, the quantity of ZnCl_2 is 40 g.

4. In order to illustrate the advantage of 3D porous VN SERS substrate, authors could provide a systematic comparison (e.g. Table) of their work vs. earlier publications.

Reply: This is a very good suggestion. By calculation, the Raman enhancement factor (EF) of the 3D porous VN substrate to R6G molecules is 5.2×10^7 . According to this suggestion, a performance comparison list (Supplementary Table 1) has been added to the Supplementary Information.

Page 19 (yellow part): By calculation, the Raman enhancement factor (EF) of the

3D porous VN substrate to R6G molecules is 5.2×10^7 , which is outstanding among the non-noble metal SERS substrates (Supplementary Table 1).

Supplementary Table 1: some of the previously reported EFs for typical non-noble Metal Raman substrate materials

Substrate	Probe molecule	Excited wavelength (nm)	Author	EF	Stability
TiO ₂ photonic microarray	MB	532	D. Qi et al. ¹	2×10^4	stable
CdTe nanoparticles	4-Mpy	514.5	Y. F. Wang et al. ²	10^4	Liable to oxidation and corrosion
ZnO nanoparticles	D266	488	H. Wen et al. ³	50	Liable to corrosion
CdS nanoparticles	4-Mpy	514.5	Y. F. Wang et al. ⁴	10^2	Liable to oxidation and corrosion
α - Fe ₂ O ₃	4-Mpy	514.5	X. Q. Fu et al. ⁵	2.7×10^4	Liable to corrosion
Cu ₂ O	4-MBA	488	L. Jiang et al. ⁶	10^5	Liable to oxidation and corrosion
CuO nanoplates	4-Mpy	514.5	Y. Wang et al. ⁷	10^2	Liable to corrosion
W ₁₈ O ₄₉ nanorods	R6G	532.8	S. Cong et al. ⁸	3.4×10^5	Liable to oxidation
Cu ₂ O superstructure	R6G	532	L. Guo et al. ⁹	8×10^5	Liable to oxidation and corrosion
MoO ₂ nanodumbbell	R6G	532	G. C. Xi et al. ¹⁰	1.2×10^6	stable
MoS ₂ nanosheet	R6G	532.8	Zheng, Z. H. et al. ¹¹	1.6×10^5	Liable to oxidation

MOF	R6G	R6G 532.8	Sun, H. Z. et al. ¹²	10 ⁶	Liabile to oxidation and corrosion
Organic Semiconductor	DFH-4T	532	Yilmaz M. et al. ¹³	3.4×10 ³	Liabile to oxidation and corrosion
Nb ₂ O ₅ nanorods	MB	532	Shan, Y. F. et al. ¹⁴	7.1×10 ⁶	stable
Amorphous ZnO Nanocages	4-MBA	633	Wang, X. T. et al. ¹⁵	6.6×10 ⁵	Liabile to corrosion
Amorphous TiO ₂ nanosheets	4-MBA	633	Wang, X. T. et al. ¹⁶	1.8 × 10 ⁶	stable
VN (the present work)	R6G	532	Guan, H. M. et al.	5.2×10⁷	stable

5. The second paragraph of Page 18 is the conclusion of this work, but the subheading was written as “Discussion”. Please revise it.

Reply: We thank the reviewers for the suggestion. We refer to other paper formats published by *Nature Communications* and found that many Conclusion are written under DISCUSSION. There are also some papers that do not list DISCUSSION, and they put their conclusions directly after the Results. Based on the reviewers’ suggestion, we now put the conclusion directly after the results. Of course, if the editor puts forward more specific format requirements, we will make further changes.

Reviewer-3's specific comments

The manuscript describes a new method for the synthesis of porous metal nitrides using ZnCl₂ as a template. The authors attribute superhydrophobic properties, adsorption of organic compounds, photo thermal conversion and SERS activity to the

transition metal nitrides synthesized by this route. The emphasis is on the new route of synthesis, however, purity and phase confirmation of synthesized metal nitrides is not proven unambiguously. Experimental details are lacking, particularly on the proposed applications.

1. Given that several applications are included in the manuscript, introduction can be a bit more elaborate including the adsorption capacities, photo thermal conversion and SERS activity of reported metal nitrides.

Reply: We are very grateful to the reviewers for this suggestion, which are very helpful in improving the quality of this paper. According to this suggestion, the relative content has been added into the revised paper.

Page 3 (yellow part): For examples, two-dimensional Ti₂N nanosheets have been reported to be an excellent SERS substrate.¹¹ Hussain et al. demonstrated that plasmonic TiN is an efficient hybrid photodetector under low light conditions.¹² More recently, Schramke et al. reported that plasmonic TMN nanocrystals with resonances at near-Infrared wavelengths have strong photothermal effects.¹³

(11) Soundiraraju, B. & George, B. K. Two-dimensional titanium nitride (Ti₂N) MXene: synthesis, characterization, and potential application as surface-enhanced Raman scattering substrate. *ACS Nano* **11**, 8892-8900 (2017).

(12) Hussain, A. A.; Sharma, B.; Barman, T. & Pal, A. R. Self-powered broadband photodetector using plasmonic titanium nitride. *ACS Appl. Mater. Interfaces* **8**, 4258-4265 (2016).

(13) Schramke, K. S.; Qin, Y.; Held, J. T.; Mkhoyan, K. A. & Kortshagen, U. R.

Nonthermal plasma synthesis of titanium nitride nanocrystals with plasmon resonances at near-infrared wavelengths relevant to photothermal therapy. *ACS Appl. Nano Mater.* **1**, 2869-2876 (2018).

2. Atomic ratio of VN (and it is calculated only for VN, why not for other nitrides also) is calculated from EDS spectrum (Figure 11), but in EDS spectrum there is an overlap of V L alpha(0.51 keV) with O K alpha (0.52 keV), hence elemental composition shall be further confirmed with some other technique like XPS.

Reply: We are very grateful to the reviewers for this suggestion. According to this suggestion, the EDS spectra of other TMNs (MoN, WN, and TiN) have been added into the revised paper. At the same time, their components are also obtained by calculating the peak area.

Page 8 (yellow part): In addition, the EDS spectra of the samples show that the atomic ratio of Mo/N, W/N, and Ti/N are about 1.08, 1.05, and 1.04 (Supplementary Figure 9).

Supplementary Figure 9. EDS spectrum of the as-synthesized 3D porous MoN, WN, and TiN.

We are also grateful to the reviewers for their suggestions to confirm the composition with XPS. However, after discussing with experts who do XPS, we found that although XPS is a very good surface component characterization method, it is not very suitable for determining the overall composition of the sample. Although XPS is an accurate surface analysis technology, its detection depth is only 2-3 nm, while the detection depth of EDS is as high as 3-5 μm . Since the current sample size is in the micron level, so it is more reasonable to use EDS to characterize its comprehensive composition.

As the reviewer said, the signals of oxygen and vanadium have some overlap. In order to further confirm the composition of the VN samples, we adopted the latest X-ray fluorescence (XRF, ZETIUM) technology. The characterization results showed that the atomic ratio of V/N is 1.09, which is basically consistent with their EDS characterization results. The relative contents have been added into the revised paper.

Page 8 (yellow part): In addition, since the signals of oxygen (O K, 0.52 KeV) and vanadium (V L, 0.51 KeV) have some overlap, in order to further confirm the composition of the sample, the X-ray fluorescence (XRF) component characterization technology is adopted, which showed that the atomic ratio of V/N is 1.09, which is highly consistent with their EDS characterization results.

3. Though the phase of the material is confirmed by XRD, presence of oxygen cannot be ruled out, particularly when ZnO is formed as intermediate during the synthesis, which is further removed by washing with 3M HCl. How will you explain the V⁴⁺ and V⁵⁺ ions detected in the XPS spectrum of VN (which is not a small amount as claimed by the authors)? XPS spectrum of other metal nitrides (Fig.S8) also shows the presence of metal oxides. Estimate elemental composition (of all elements) using XPS.

Reply: We are very grateful to the reviewer for this question. Below we will explain this question carefully. First, the XRD characterization results show that these transition metal nitrides are highly pure, and no metal oxide phases are detected. Of course, this does not mean that there is no metal oxide in the samples, because the detection limit of XRD (Bruker D8 Focus) is about 0.5-1% (mass ratio). The XRD results can at least prove that most of the samples (greater than 99%) are transition metal nitrides. The reason why there are more oxide signals on the XPS spectra of the TMNs is that XPS is an extremely sensitive surface composition analysis instrument. The detection depth of XPS on the sample is only 2-3 nm, and it is difficult to avoid some oxides on the surface of the sample, therefore, there will be some metal oxide

signals in the XPS signal. Moreover, by investigating the literature, we found that this is a common phenomenon in TMN synthesis (*Adv. Funct. Mater.* 2018, **28**, 1805510; *ACS Nano* 2017, **11**, 2180-2186).

Adv. Funct. Mater. 2018, **28**, 1805510

ACS Nano 2017, **11**, 2180-2186

4. How the complete removal of ZnO is ensured during the washing step?

Reply: We are very grateful to the reviewer for this question. ZnO is an easily soluble substance, and its solubility in dilute hydrochloric acid is extremely high. After pickling, no ZnO was found in the XRD pattern (Supplementary Figure 1). There is also no signal of zinc in the EDS spectrum (Figure 11). Moreover, according to the suggestion of the reviewer, in order to further confirm that there is no zinc oxide, we analyzed the sample by ICP-MS, and the results showed that there was still no signal of zinc ion. These results demonstrated that ZnO has been completely eliminated during the pickling process.

5. The proposed formation mechanism (Fig.3j) is confusing. The picture gives the impression of ZnO particles in MN framework.

Reply: We are very grateful to the reviewers for pointing out the problems in this mechanism diagram. According to this suggestion, we redraw the formation mechanism diagram, hoping to explain the formation process more clearly.

Figure 3 (j). Formation mechanism scheme of the 3D porous TMNs.

6. Attributing superhydrophobic properties to the TMNs is not correct as all the synthesized metal nitrides have contact angle in the range of 107° to 128° . A material is said to be superhydrophobic, when the contact angle is more than 150° .

Reply: We are very grateful to the reviewers for pointing out our incorrect statement.

In the revised version, all “superhydrophobics” have been replaced with “hydrophobic”.

7. Experimental details of adsorbability, photothermal effect and hydrophobicity studies?

Reply: We are very grateful to the reviewer for these suggestions. The Experimental details of adsorbability, photothermal effect and hydrophobicity studies have been added into the Supporting Information.

Page 1-2 in Supporting Information (yellow part):

Adsorption experiment

During the adsorption test, 50 mg of the 3D porous TMNs were added into the solvent to be adsorbed (100 mL) at room temperature and one atmosphere. After 10 min of full contact, the 3D porous TMNs were quickly separated by vacuum filtration.

Weight measurements should be made as soon as possible to avoid evaporation of organic liquids with low boiling points. The weight of 3D porous TMNs before and after adsorption was recorded, and the weight increment was calculated.

Photothermal Test

50 mg of 3D porous VN was dispersed well in 10 mL distilled water, under the assistance of ultrasonic bath. The dispersed mixture was deposited on the foam rubber membrane under vacuum. The formed VN/foam rubber film was placed on a heat plate and the temperature was kept at 70 °C for 12 min. Foam rubber was chosen as a bottom supporting layer because of its unique inner microporous structure and hydrophilicity. The microporous structure of the cellulose membrane enables efficient absorption of water through capillary effect. This effect enables more rapid replenishment of surface water after evaporation, while the hydrophilicity would benefit the water adhesion and speed up the water transfer upward. Under illumination of a solar simulator at power density of 2 KW m⁻², the light-thermal system can be quickly heated up in 10 s and generates visible steam flow on top of the water surface.

Hydrophobicity Measurement

The contact angle was measured using a contact - angle measurement system (Contact Angle System OCA 20, Dataphysics). For the test of hydrophobic property, a water droplet was placed on the top by a syringe needle. The droplet keeps a round shape and merges into the structure. The picture of the formed droplet was taken by

an optical microscope from the side view and was then inserted into an “Image Software” . The contact angle was calculated by the Software for different substrates.

8. In Fig. S14 caption and label in the figure do not match

Reply: We are very sorry for this typo. The caption of the Supplementary Figure 17 (in revised Supporting Information) have been corrected.

Supplementary Figure 17. Saturated adsorption capacity of TiN for organic compounds.

9. The stability of VN to harsh environment, heat and corrosion is mainly characterized by V2p XPS spectrum (Figure 5i). It is better to include XPS survey spectra and corresponding N1s and O1s spectra also to support the claim.

Reply: We are very grateful to the reviewer for these suggestions. The XPS survey spectra and corresponding N1s and O1s spectra are provided in the revised Supporting Information (Supplementary Figure 18).

Supplementary Figure 18. XPS characterization of the VN samples after various treatment. (a-c) Survey spectra. (d-f) N1s spectra. (g-i) O1s spectra. It can be seen from these spectra that the surface states of these VN samples remain almost unchanged after various treatments. It should be noted that the clear O1s line confirms that a thin oxide layer exists on the surface of VN. The oxygen signal is fitted with two peaks. The main component is centered at 530.3 eV and is typical for oxygen in a metal oxide. The second small peak is at about 531.7 eV and can be attributed to the signal from -OH groups chemisorbed at the surface.

10. The authors claim 'high' thermal stability for the TMNs. The experimental data provided shows thermal stability only up to about 300°C. Include a TGA under inert atmosphere to support the claim of stability up to 800°C.

Reply: We are very grateful to the reviewer for this question. in the 300 °C TGA experiment, the atmosphere used is air, mainly to investigate the oxidation resistance

of VN. Considering the high specific surface area and high porosity of these 3D porous VN, their oxidation resistance is excellent. Heating to 800 °C in nitrogen is to investigate the thermal stability of the 3D porous VN, which has been demonstrated by XPS (Figure 5i and Supplementary Figure 18). According to the suggestion of the reviewer, the TGA data obtained at 800 °C under N₂ atmosphere has been added into the revised Supporting Information (*Supplementary Figure 19*), which also demonstrated the high thermal stability of the VN.

11. The weight increase as shown in the TG in air for VN (Fig.5a), is not matching with theoretical value (this inference is only based on the figure provided, quantitative data is not available). This supports my earlier observation that the VN phase is not pure. Please do a quantitative estimate of weight increase due to oxidation and compare it with theoretical value

Reply: We are very grateful to the reviewers for pointing out this unusual TG data. I checked the TG data myself. It is true that, as the reviewer said, the added quality is not enough (Theoretically, when VN of 2.1 g is completely converted to V_2O_5 , its mass should be 2.9 g instead of the current 2.55 g). I think this is mainly due to the errors introduced by my students during the experiment. I have arranged for experienced professionals to redo the experiment, and the results are in line with the expected values. The relevant data has been put in the revised article. Of course, as the reviewer said, due to some vanadium oxide on the surface of the porous VN, the added mass is still slightly smaller than the theoretical value, which is reasonable.

Figure 5. Thermal, chemical and light stability of the 3D porous VN. (a) TG curve.

12. To measure SERS activity, enhancement factor of VN for model compound is to be included in the manuscript.

Reply: We are very grateful to the reviewer for this good advice. The calculation of Raman enhancement factor (EF, 5.2×10^7 , page 18 of the main text) of VN has been done, and the relevant experiments and calculation process have been added in the Supporting Information (page 2).

Enhanced Factor Calculation

To calculate the EF of the 3D porous VN samples, the ratio of SERS to normal Raman spectra (NRS) of R6G was determined by using the following calculating

Formula 1

$$EF = (I_{SERS}/N_{SERS})/(I_{NRS}/N_{NRS}) \quad (1)$$

$$N_{SERS} = N_A n S_{Irr}/S_{dif} \quad (2)$$

$$N_{NRS} = d S_{Irr} h N_A / M \quad (3)$$

where I_{SERS} and I_{NRS} refer to the peak intensities of the SERS and NRS, respectively.

N_{SERS} and N_{NRS} correspond to the number of probe molecules excited in the SERS and

NRS tests. In the SERS measurements, two Raman scattering peaks, R_1 at 612 cm^{-3}

and R_2 at 773 cm^{-3} were selected for the calculations of the EF. For comparison, the

peak intensities of the R6G ($1 \times 10^{-2} \text{ M}$, aqueous solution) directly placed on bare

glass were detected as NRS data. To decrease the measuring error, the intensities were

obtained by continually ran the test procedure at randomly selected 10 points and took

the average. N_{SERS} is calculated by formula 2, where N_A refer to the Avogadro's

constant, n correspond to the molar quantity of the probe molecule, S_{Irr} refer to the

irradiation area under the laser beam ($5 \mu\text{m}$ in diameter), and S_{dif} refer to the diffusion

area of the substance to be tested on the substrate. In a typical test, one drop (20

microliter) of the probe solution was dropped onto the SERS substrate, and the probe

solution was spread into a circle with a diameter of 4 mm when the solution is

completely dry. N_{NRS} is determined by the formula 3, where d is the packing density

of R6G molecules in the surface of substrate ($1.4 \times 10^{21} \text{ molecule/cm}^3$), h refer to the

laser confocal depth ($26 \mu\text{m}$), M correspond to the molecule weight of R6G (479).

13. The SERS sensitivity of 10^{-11} M is questionable. The experiment includes dipping of VN coated glass plate into the sample solution. What is the quantity of VN on glass plate and what is the quantity of the solution used? According to the authors, 'VN microparticles have extremely high adsorption capacity for DCP'. Therefore, the enhancement shown in Raman spectrum could be due to the extremely high concentration of DCP on VN particles. This is also evident from Fig.7a, where all the R6G is detected on VN and not even a trace is detected on the glass surface. Also, the authors report a beam spot size of 5 mm for the Raman measurement, which is unusual.

Reply: We are very grateful to the reviewer for this good advice. According to the reviewer's suggestion, detailed experimental parameters of SERS test are added in the revised paper. Indeed, as the reviewer said, the excellent SERS performance of these VN porous particles is closely related to their large surface area and extremely high adsorption capacity. In addition, the beam spot size of 5 mm is a typing error, actually it is 5 μm .

14. Figure 7d presents LDL (y-axis) vs treatment conditions (x-axis), Raman spectra to be included to show the detection of PCPs and high stability of VN substrate as claimed by the authors.

Reply: This is a good idea. Following the reviewer's recommendations, we added the Raman spectra of the 2,5-DCP into the Supporting Information (Supplementary Figure 22).

Supplementary Figure 22. The Raman spectra of 2,5-DCP obtained on 3D porous VN after various treatments. Laser power: 0.5 mW; Integration time: 60 s.

15. IR and Mass spectrometry results are not discussed in manuscript but included in experimental section

Reply: We are very sorry for this mistake. The *Characterization* in experimental section is written in the usual way without careful examination. We are very sorry for this low-level mistake. The IR has been deleted from the experimental section.

16. Out of curiosity, have the authors attempted desorption of adsorbed organic species.

Reply: This is a very good idea, and we are very grateful to the reviewers for this suggestion. However, how to design and operate these desorption experiments is a very complicated process, which involves many factors such as desorption temperature, pressure and vacuum degree, so it is not a short time to get a more accurate conclusion. But this is really a very good suggestion. We plan to study this issue as a new topic and hope to answer this question in the near future. Thanks again.

17. Typographic errors to be corrected and grammar check required

Reply: We are very grateful for the reviewer's suggestion. We have carefully checked

the full text and tried our best to correct grammatical and typographical errors.

REVIEWERS' COMMENTS

Reviewer #1 (Remarks to the Author):

It is suggested to be accepted now.

Reviewer #2 (Remarks to the Author):

The general molten salt preparation method for porous TMN materials reported in this work was quite novel and interesting. The authors also demonstrated the application as SERS substrates. The manuscript was improved in the revised version and the reviewers' concern have satisfactorily addressed. The manuscript can be accepted for publication in Nature Communications.

Reviewer #3 (Remarks to the Author):

The authors have incorporated most of the suggestions made by the reviewer in the revised manuscript and wherever it is not done, sufficient explanations are provided. The revised manuscript makes abetter reading and factual errors were corrected. However, the explanation provided by the authors for avoiding XPS analysis to determine the atomic ratios of metal nitrides is not acceptable. Except for this, all other queries were sufficiently explained and the revised manuscript is modified accordingly

The manuscript may be cleared for publication